# Ultrathin metal–organic framework membrane production by gel–vapour deposition

Wanbin Li[1], Pengcheng Su[2], Zhanjun Li[1], Zehai Xu[2], Fei Wang[1], Huase Ou[1], Jiaheng Zhang[2], Guoliang Zhang[2] & Eddy Zeng[1]

Ultrathin, molecular sieving membranes composed of microporous materials offer great potential to realize high permeances and selectivities in separation applications, but strategies for their production have remained a challenge. Here we show a route for the scalable production of nanometre-thick metal–organic framework (MOF) molecular sieving membranes, specifically via gel–vapour deposition, which combines sol–gel coating with vapour deposition for solvent-/modification-free and precursor-/time-saving synthesis. The uniform MOF membranes thus prepared have controllable thicknesses, down to ~17 nm, and show one to three orders of magnitude higher gas permeances than those of conventional membranes, up to $215.4 \times 10^{-7}$ mol m$^{-2}$ s$^{-1}$ Pa$^{-1}$ for $H_2$, and $H_2/C_3H_8$, $CO_2/C_3H_8$ and $C_3H_6/C_3H_8$ selectivities of as high as 3,400, 1,030 and 70, respectively. We further demonstrate the in situ scale-up processing of a MOF membrane module (30 polymeric hollow fibres with membrane area of 340 cm$^2$) without deterioration in selectivity.

[1] School of Environment, Guangzhou Key Laboratory of Environmental Exposure and Health, and Guangdong Key Laboratory of Environmental Pollution and Health, Jinan University, Guangzhou 510632, People's Republic of China. [2] Institute of Oceanic and Environmental Chemical Engineering, State Key Lab Breeding Base of Green Chemical Synthesis Technology, Zhejiang University of Technology, Hangzhou 310014, People's Republic of China. Correspondence and requests for materials should be addressed to W.L. (email: gandeylin@126.com) or to G.Z. (email: guoliangz@zjut.edu.cn)

Membrane separation is a more efficient and environmentally friendly technology than traditional methods based on adsorption, distillation and extraction. Molecular sieving membranes composed of microporous materials, such as carbons[1,2], zeolites[3,4] and metal–organic frameworks (MOFs)[5,6] are tremendously beneficial in permeability and selectivity compared with polymeric membranes. The membrane flux is usually inversely correlated to the thickness of dense layer[7]. To ensure the continuity for achieving high selectivity, molecular sieving membranes are often formed with thick dense layers from several micrometres to tens of micrometres to eliminate the effects of superposed crystallization[8]. Micrometre-thick MOF composite membranes with high selectivity have been fabricated by transformation, yet the gas permeances are moderate[9]. Ultrathin MOF films are deposited on dense substrates for electronic, optical and sensing applications, through chemical vapour deposition and layer-by-layer synthesis[10–14]. However, molecular sieving MOF membranes for separation applications require stringent membrane continuity. The growth mechanism for the formation of MOF membranes on porous substrates is also vastly different from that of MOF films on dense substrates. MOF membranes can be prepared with various methods[5,8], such as hydro/solvothermal synthesis[15–18], direct crystallization[19–21] and interfacial growth[22–24], e.g., micrometre-thick MOF membranes were synthesized on hollow fibres via interfacial microfluidic membrane processes[23,24]. Metal-rich coatings can also be employed to fabricate MOF membranes as buffering/seed layers[18,25–30], but the coating layers are often prepared via complex procedures and a second hydro/solvothermal treatment is necessary for obtaining continuous MOF membranes. So far, preparation of molecular sieving membranes with nanometre-thick dense layers or in a scalable route has remained a great challenge[5,6,23,31]. The complex reactant transport, heterogeneous crystallization and fluid dynamics should be controlled.

We report a gel–vapour deposition (GVD) methodology for the production of ultrathin MOF membranes in a scalable and environmentally friendly manner, by combination of modification-free sol–gel coating and solvent-free vapour deposition. Through GVD, the thickness of the MOF membranes is easily controllable. A nanometre-thick MOF membrane (<20 nm) can be obtained by adjustment of sol concentrations and coating procedures. Some of the benefits with GVD include no need for pretreatment of substrates and solvents during crystallization, excellent compatibility between MOF layers and substrates, reuse of expensive MOF precursors, easy manipulation of the positions of MOF layers and time-effective synthesis processes. Use of GVD allows in situ (and small-scale) preparation of MOF membranes in a module with large effective membrane areas, and can fabricate MOF membranes with different topologies on multifarious substrates. ZIF-8, a MOF consisting of zinc centres and 2-methylimidazole ligands, shows great promise in $H_2$ recovery, $CO_2$ capture and hydrocarbon separation[32]. Polymeric hollow fibres have the benefits of low cost, large membrane area per volume and commendable processing ability[33]. Herein we exhibit the utility and distinctive features of GVD by deposition of ZIF-8 on polymeric hollow fibres.

## Results

**Fabrication of ZIF-8 membranes by GVD.** To fabricate the ultrathin ZIF-8 membranes by GVD, Zn-based sol prepared by mixing zinc acetate dihydrate and ethanolamine in ethanol was coated on ammoniated polyvinylidene fluoride (PVDF) hollow fibres, and heat treated to form Zn-based gel[34]. The hollow fibers were ammoniated by ethanediamine to improve the stability as previously reported[35]. The gel layer was transformed to the MOF membrane directly through ligand vapour deposition by heat treatment (Fig. 1a). In the process of MOF crystallization, the

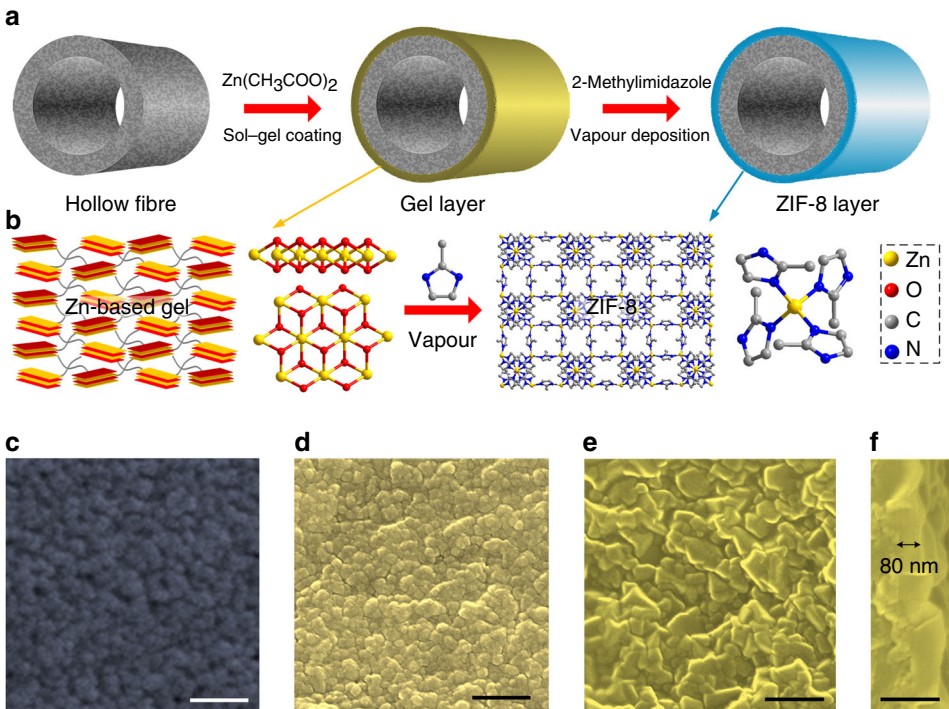

**Fig. 1** GVD fabrication of ultrathin ZIF-8 membrane. **a** Schematic of MOF membrane formation process. **b** Schematic illustration and chemical structure of Zn-based gel and crystalline structure of ZIF-8. Zn, O, C and N atoms are depicted in *yellow*, *red*, *grey* and *blue*, respectively. H atoms are not presented for clarity. Top view SEM images of **c** the PVDF hollow fibre and **d** the Zn-based gel layer. SEM images of **e** top and **f** cross-sectional view of the ZIF-8 membrane prepared with sol concentration of 1 U and coating time of 2 s. The images are coloured for clarity. *Scale bar*, 200 nm

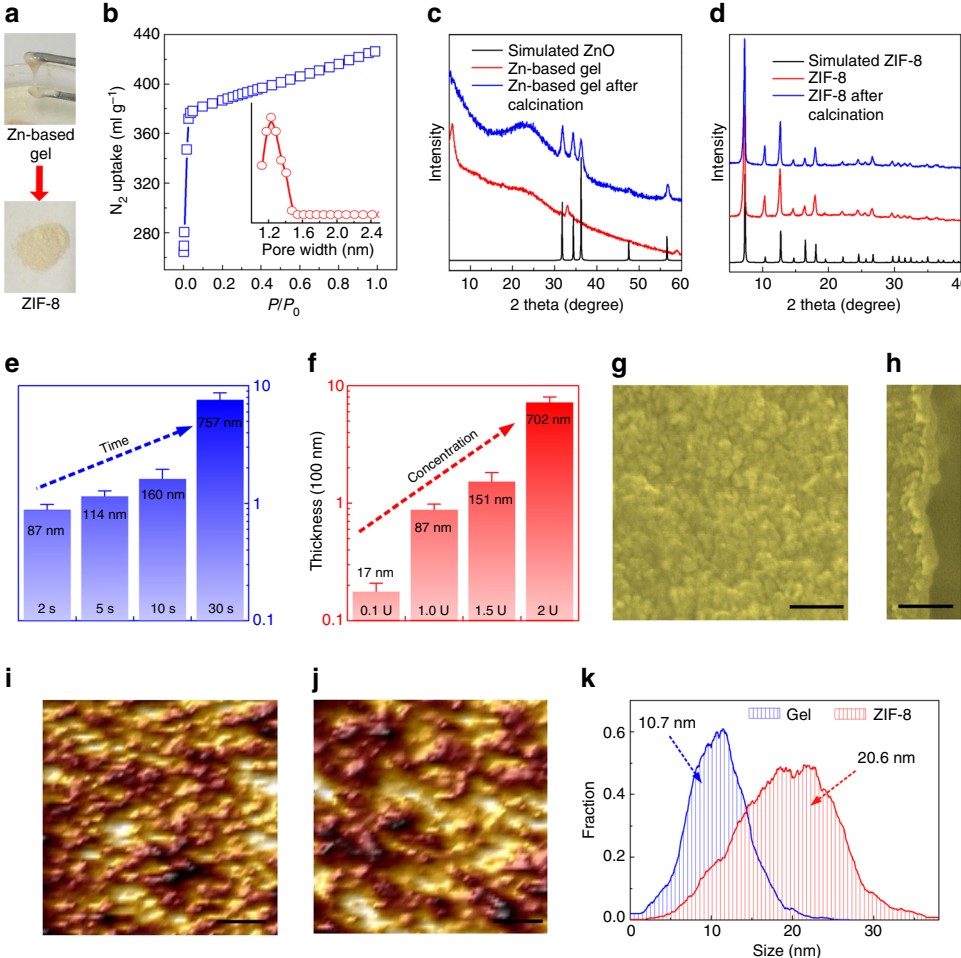

**Fig. 2** Characterizations and thickness of ZIF-8 membranes. **a** The optical photographs of the Zn-based gel and the ZIF-8 powder. **b** $N_2$ adsorption isotherm and corresponding pore width distribution of the prepared ZIF-8. **c** XRD patterns of the simulated ZnO, the Zn-based gel layer and the calcinated Zn-based gel layer on PVDF hollow fibres. **d** XRD patterns of the simulated ZIF-8, the ZIF-8 membrane and the calcinated ZIF-8 membrane. The samples were prepared with sol concentration of 1 U, coating time of 2 s and vapour deposition time of 2 h. Thickness of ZIF-8 membranes prepared with different **e** coating time and **f** sol concentration. The thickness was measured at various locations to obtain the standard deviation. **g** Top and **h** cross-sectional view SEM images of the ZIF-8 membrane. *Scale bar*, 200 nm. AFM analysis of **i** the Zn-based gel layer and **j** the ZIF-8 membrane. $R_q$: **i** 3.6 nm and **j** 5.8 nm. $R_a$: **i** 2.8 nm and **j** 4.7 nm. *Scale bar*, 100 nm. **k** Grain size distribution of the Zn-based gel layer and the ZIF-8 membrane from AFM data. The samples were prepared with sol concentration of 0.1 U, coating time of 2 s and vapour deposition time of 2 h

produced vapour interacted with the sensitive Zn-based gel and substituted the organic chains to form more stable ZIF-8 (Fig. 1b). Scanning electron microscopy (SEM) images of the prepared membranes (Fig. 1c–f) show an impermeable dense gel layer coated on the porous substrate, which is also verified by gas permeation measurement. After deposition, a continuous and uniform ZIF-8 layer was deposited on the surface of the hollow fibre (Supplementary Fig. 1). The crystals are nanosized and well intergrown. The process of GVD completely avoids the swelling of polymeric hollow fibres, thus preventing the formation of cracks in the prepared MOF membranes. The difference of swelling between MOF layers and substrates would lead to formation of shear forces, and therefore cracks. At the same time, the consumption of expensive MOF precursors is greatly reduced, as a result of reusable precursors and no bulk MOF growth. The cross-sectional SEM image (Fig. 1f) shows that some ZIF-8 crystals are injected into the dense layer of the hollow fibre, which can greatly enhance the adhension between the dense layers and substrates. The thickness of the ZIF-8 layer is ~87 nm, the lowest value achieved so far. Benefitted from the simple coating process, the position of MOF layers is manipulated. We can also deposit a

continuous ZIF-8 layer on the inner surface of the hollow fibre (Supplementary Fig. 1). Outer-surface membranes provide larger effective area than inner-surface membranes, but inner-surface membranes are likely to show greater mechanical stability. Because the metal source position is immobilized and the phases of two precursors are different, crystallizaton is confined as a gas–solid reaction. The process does not involve the diffusion of precursors in solutions, and therefore the reactant transport and fluid dynamic processes during crystallization can be greatly controlled.

We synthesized ZIF-8 powders through the same procedures as GVD to characterize their properties and crystallizaton mechanisms. It is clear that the viscous Zn-based gels have been transformed to granular ZIF-8 materials after deposition (Fig. 2a). The broad characteristic peaks in the powder X-ray diffraction (XRD) pattern of Zn-based gel indicate the existence of tiny layered basic zinc acetate crystals (Supplementary Fig. 2). This can be explained by the crystallization between zinc ions and bonding water from zinc acetate dihydrate in alkaline ethanolamine solution. Besides promoting crystallization of layered basic zinc acetate, ethanolamine can also react with zinc acetate to form

complexes[36]. The gel network structure is composed of tiny layered basic zinc acetate components and ethanolamine-zinc complexes. In deposition, 2-methylimidazole coordinates with zinc to form ZIF-8. The substituted organic groups from gels capture protons from 2-methylimidazole to become small organic molecules, and are evaporated upon thermal treatment. XRD patterns verify that the coating layer is transformed from gel to definite ZIF-8[37, 38]. Fourier transform infrared (FTIR) spectra reveal that the OH ($3,000–3,700 \, cm^{-1}$), $CH_2$ (2,941 and $2,885 \, cm^{-1}$) and C=O (1,581 and $1,401 \, cm^{-1}$) groups of the Zn-based gel disappear after deposition, whereas the N-Zn ($420 \, cm^{-1}$), $CH_3$ ($3,136 \, cm^{-1}$) and CH ($2,931 \, cm^{-1}$) characteristic peaks of ZIF-8 appear (Supplementary Fig. 3)[39]. $N_2$ adsorption isotherm confirms that the resulting ZIF-8 possesses remarkable porosity with Brunauer–Emmett–Teller (BET) surface area of $1,658 \, m^2 \, g^{-1}$ (Fig. 2b), which is similar to those of the ZIF-8 materials synthesized by the solvothermal methods[37, 38]. Pore size distribution suggests that the ZIF-8 possesses pore width of ~1.2 nm due to the Sodalite (SOD) cage.

**Controlling of membrane fabrication.** The effect of deposition time on membrane formation was investigated. The vapour deposition time was set as 30, 120 and 360 min. XRD characterization confirms that all three samples show intact ZIF-8 crystalline structure. Zn-based gel can be mineralized to ZnO by calcination at 200 °C (Fig. 2c). To examine whether gels were completely transformed to ZIF-8, three samples with different deposition times were calcined. XRD patterns show almost no change in crystalline structures and no characteristic peaks of ZnO (Fig. 2d and Supplementary Fig. 4), suggesting that all Zn-based gels were transformed to ZIF-8, even with a

crystallization time of 30 min only. This also verifies the excellent thermal stability of the prepared membranes. Rapid crystallization is resulted from the high activity of gel promoting substitution reaction and the evaporation of substituted materials facilitating ZIF-8 formation. To ensure the continuity of membranes, commonly used MOF growth processes usually need to last for several hours to several days in traditional hydro/solvothermal methods. Obviously, GVD can greatly shorten the time for synthesizing MOF membranes.

The thickness of ZIF-8 membranes can be easily manipulated by controlling coating time and sol concentration. Although greater thickness usually leads to better continuity, smaller thickness means higher permeance. The thickness of ZIF-8 layer varied with coating time and sol concentration (Fig. 2e, f). For convenience of discussions, $1/3 \, g \, ml^{-1}$ of zinc acetate dihydrate/ethanol sol was defined as one concentration unit (U) in the present study. When sol concentration was 1 U, the membrane thickness increased from 87 to 757 nm with coating time extending from 2 to 30 s due to increased gel loading (Fig. 2e and Supplementary Fig. 5). The thickness of membranes with the coating time of 2 s decreased with decreasing sol concentration (Fig. 2f and Supplementary Fig. 6). When the Zn-based sol concentration was further diluted to 0.05 U (diluting the zinc acetate dihydrate/ethanol sol of $1/3 \, g \, ml^{-1}$ by 20 times), some pinholes were formed in the membrane. This can be attributed to the high porosity of substrates and low viscosity of sol. Low viscosity led to small loading and fast spread of Zn-based sol in porous substrates (Supplementary Fig. 7). It is noteworthy that the membrane prepared with the coating time of 2 s and concentration of 0.1 U (diluting the zinc acetate dihydrate/ethanol sol of $1/3 \, g \, ml^{-1}$ by 10 times) is well intergrown and defect free, and has a thin thickness of ~17 nm (Fig. 2g, h).

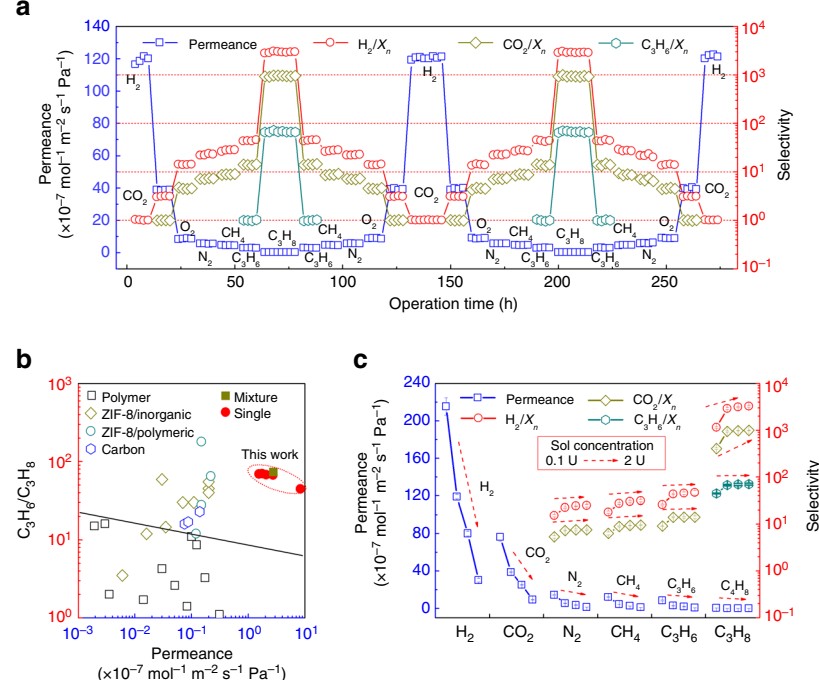

**Fig. 3** Gas transport behaviours of ZIF-8 membranes. **a** Single gas transport behaviours through the ZIF-8 membrane prepared with sol concentration of 1 U, coating time of 2 s and vapour deposition time of 2 h. $X_n$ represents various gases. Permeation data were collected for two cycles over 274 h, from smallest $H_2$ with kinetic diameter of 0.289 nm to largest $C_3H_8$ with kinetic diameter of 0.43 nm and then in reverse order. **b** Comparison of ZIF-8 membranes here with polymeric[50], carbon[47] and other MOF membranes in literatures for $C_3H_6/C_3H_8$ system. The *black line* is the upper bound of polymeric membranes[50]. The numerical data are shown in Supplementary Table 2. The permeance is calculated from the permeability by assuming membrane thickness of 1 μm. **c** Single gas permeation behaviours of different gas through ZIF-8 membranes prepared with various sol concentrations of 0.1, 1, 1.5 and 2 U. These ZIF-8 membranes were prepared with coating time of 2 s and vapour deposition time of 2 h

The thickness is ~60–1,000 times smaller than those obtained by other previous studies (Supplementary Table 1). Atomic force microscopy (AFM) images indicate that the roughness of the membrane has increased after deposition, apparently attributed to the formation of polyhedral crystal structures (Fig. 2i, j and Supplementary Fig. 8). However, the arithmetic average roughness ($R_a$) and root mean square roughness ($R_q$) of the ZIF-8 membrane prepared with a concentration of 0.1 U are only 4.7 and 5.8 nm, respectively, calculated over a probe area of 500 × 500 nm$^2$ area. This roughness is lower than that of the smoothest ZIF-8 film available[10], and greatly lower than those of MOF membranes[5, 8]. The crystal size of ZIF-8 obtained by AFM is 20.6 nm (Fig. 2k), consistent with membrane thickness.

**Gas transport behaviours of ZIF-8 membranes.** We characterized the permeance of various gases through ultrathin ZIF-8 membranes (Fig. 3a). Gas permeances remained largely intact after two measurement cycles of 274 h, demonstrating the extraordinary stability of ZIF-8 membrane. The permeance of $H_2$ through the ZIF-8 membrane was much higher than those of other gases. The selectivities of $H_2/O_2$, $H_2/N_2$, $H_2/CH_4$, $H_2/C_3H_6$ and $H_2/C_3H_8$ reached 14.1, 22.4, 27.3, 43.1 and 2,894, respectively, at the first half cycle. These values are superior to Knudsen diffusion coefficients (4.0, 3.7, 2.8, 4.6 and 4.7) and also higher than most of those obtained from previous studies (Supplementary Table 1). We also tested the gas permeation of uncoated PVDF hollow fibres, obtaining a much lower value (3.1) of $H_2/C_3H_8$ selectivity. These suggest that the ZIF-8 layer is defect free and plays a critical role in good permselectivity of the membrane. The ultrathin ZIF-8 membranes thus prepared possess one to three orders of magnitude higher $H_2$ permeance (119.0 × 10$^{-7}$ mol m$^{-2}$ s$^{-1}$ Pa$^{-1}$) than those of zeolite, polymers of intrinsic microporosity and MOF membranes reported in the literatures (Supplementary Table 1). The efficiency of our ZIF-8 membranes for $H_2$ separation easily exceeds the Robeson upper-bound of polymeric membranes (Supplementary Fig. 9)[40]. Besides competitive $H_2$ permselectivity, the ultrathin ZIF-8 membrane also displayed a permeation cutoff between $CO_2$ (0.33 nm) and $O_2$ (0.346 nm) because the crystallographic aperture of ZIF-8 is 0.34 nm, and possessed selectivity values of 4.6, 7.3, 8.9, 14.0 and 940 for $CO_2/O_2$, $CO_2/N_2$, $CO_2/CH_4$, $CO_2/C_3H_6$ and $CO_2/C_3H_8$, respectively, which are similar to those of the ZIF-8 membranes as previously reported[41]. The $CO_2$ permeance of as high as 38.6 × 10$^{-7}$ mol m$^{-2}$ s$^{-1}$ Pa$^{-1}$ was achieved simultaneously. The framework flexibility of ZIF-8 originating from the rotation of imidazole molecules allowed gases with large kinetic diameters to pass through the ZIF-8 membranes[41, 42].

As previously demonstrated, ZIF-8 membranes have the great capability of separating $C_3H_6$ and $C_3H_8$[43, 44]. The ZIF-8 membrane prepared in the present study exhibited excellent permeance for $C_3H_6$ (2.8 × 10$^{-7}$ mol m$^{-2}$ s$^{-1}$ Pa$^{-1}$) which is one to two orders of magnitude higher than carbon and other MOF membranes[16, 23, 30, 41, 43–49], but also with outstanding $C_3H_6/C_3H_8$ selectivity at 67.2 (Fig. 3b and Supplementary Table 2). The ZIF-8 membranes also considerably outperform polymeric membranes[50], and satisfy the proposed commercial performance criteria[44]. The ZIF-8 membranes prepared under other conditions also exhibited attractive gas permeability. The gas permeance decreased with increasing coating time due to increased thickness of the ZIF-8 layer. When the 0.1 U sol was applied, a 2 s coating time was enough for obtaining defect-free membranes; the selectivities were only slightly improved with extended coating time. The selectivities of the ZIF-8 membrane for $H_2/C_3H_8$, $CO_2/C_3H_8$ and $C_3H_6/C_3H_8$ with coating time of 30 s were 3,401, 1,035 and 70.8, respectively (Supplementary Fig. 10). Upon dilution of sol concentration, the gas permeances were improved dramatically but the selectivities somewhat deteriorated (Fig. 3c). For example, the ZIF-8 membrane prepared with coating time of 2 s and concentration of 0.1 U exhibited excellent $H_2$ permeance at 215.4 × 10$^{-7}$ mol m$^{-2}$ s$^{-1}$ Pa$^{-1}$, and relativity small but still sharp $H_2/N_2$, $H_2/CH_4$, $H_2/C_3H_8$, $CO_2/N_2$, $CO_2/CH_4$, $CO_2/C_3H_8$ and $C_3H_6/C_3H_8$ selectivities at 15.1, 17.9, 1,145, 5.3, 6.3, 404 and 44.5, respectively.

We investigated the mixture separation performance of the ZIF-8 membrane prepared with the sol concentration of 1 U, coating time of 2 s and vapour deposition time of 2 h. In binary mixture separation, the ZIF-8 membrane showed the selectivities of 3,126 for $H_2/C_3H_8$ and 73.4 for $C_3H_6/C_3H_8$ (Supplementary Fig. 11). In $C_3H_6/C_3H_8$ separation, the practical pressure is usually high, and hence we further tested the effect of feed pressure on separation performance. Both the $C_3H_6/C_3H_8$ selectivity and the $C_3H_6$ permeance decreased with increasing feed pressure (Supplementary Fig. 12), similar to previous findings[30, 48, 49]. This result is different from the gas transport behaviours of polymeric membranes. Both $C_3H_6$ and $C_3H_8$ can induce plasticization of polymers. Swelling of polymers occurs in pressured feed gases, leading to the increased segmental motion and fractional free volume. Therefore, the polymeric membranes show larger gas permeability and lower selectivity under higher pressure, especially when the feed pressure exceeds the plasticization pressure[31, 51–54]. The molecular sieving MOF membranes are unaffected by plasticization. The phenomenon observed in the present study is attributed to two factors. First, more $C_3H_8$ molecules entered into ZIF-8 pores under higher pressure, which blocked the pores and formed competition[30, 31, 48], leading to the

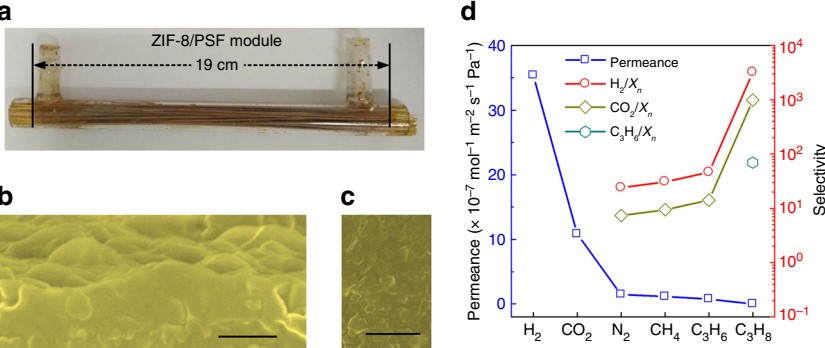

**Fig. 4** ZIF-8 hollow fibre module. **a** Optical photograph of ZIF-8 hollow fibre membrane module. SEM images of **b** top and **c** cross-sectional view of the ZIF-8 membrane in module. *Scale bar*, 400 nm (in **b**) and 2 μm (in **c**). **d** Gas permeation behaviours of various gases through the ZIF-8 hollow fibre membrane module

decreased $C_3H_6$ permeance and selectivity. Second, because $C_3H_6$ permeance was much larger than $C_3H_8$ permeance, the molar ratio of $C_3H_6/C_3H_8$ at membrane surface was smaller than that of the feed gas mixture, known as concentration polarization[55, 56]. Moreover, the increased amount of permeated $C_3H_6$ was greater than that of $C_3H_8$ with increasing pressure. The $C_3H_6/C_3H_8$ ratio at membrane surface decreased with increasing feed pressure, resulting in decreased $C_3H_6$ permeance and $C_3H_6/C_3H_8$ selectivity. Although the separation performance decreased with increasing pressure, the membrane still showed high $C_3H_6$ permeance of $2.0 \times 10^{-7}$ mol m$^{-2}$ s$^{-1}$ Pa$^{-1}$ and $C_3H_6/C_3H_8$ selectivity of 22.7 at 9 bar. These results further verify that the ZIF-8 membranes prepared by GVD have great potential in gas separation.

**In situ production of ZIF-8 membranes**. We further employed GVD for the in situ production of MOF hollow fibre membrane module directly to demonstrate the production scalability. To illustrate the general applicability and modification-free property, unmodified polysulfone (PSF) substrates (30 hollow fibres with an effective membrane area of 340 cm$^2$) were packaged in a glass shell to obtain membrane module. The 0.16 cm outer diameter of hollow fibres and 1.40 cm inner diameter of glass shell led to a high packaged fraction of 0.39 and an area per volume of 980 m$^2$ m$^{-3}$. In ZIF-8 membrane production, the Zn-based sol was impregnated into the shell side to fabricate the gel layer, and crystallization was conducted by sealing and heat-treating the ligand in containers in contact with module hermetically (Supplementary Fig. 13). The produced ligand vapour flowed into the shell side of the module and interacted with the Zn-based gel to form ZIF-8. The containers were placed at downside to promote vapour diffusion and prevent direct contact between the melted ligand and gel. The colour change of the membranes may be caused by the reaction between polymer and ethanolamine in sol (Fig. 4a and Supplementary Fig. 14). SEM images show that a crack-free ZIF-8 layer of 280 nm thickness covers on hollow fibres tightly (Fig. 2b, c). The large membrane area and high permeance can result in the high permeate side flow rate, which will lead to high concentration polarization. Based on the similar data of the single and mixture permeations, we investigated the single gas permeation behaviours of the module. The membrane module exhibited reduced gas permeances due to thicker ZIF-8 layer and denser substrate, but $H_2$ permeance remained as high as $35.4 \times 10^{-7}$ mol m$^{-2}$ s$^{-1}$ Pa$^{-1}$ and displayed $H_2/CH_4$, $H_2/C_3H_8$, $CO_2/CH_4$, $CO_2/C_3H_8$ and $C_3H_6/C_3H_8$ selectivities of 30.7, 3,212, 9.4, 983 and 69.5, respectively (Fig. 2d). This means that the ZIF-8 membranes have excellent molecular sieving properties, even if they are produced by in situ growth in module.

## Discussion

We have developed a GVD methodology for the production of nanometre-thick MOF membranes in a scalable and environmentally friendly scheme, combining sol–gel coating and vapour deposition. The GVD method can greatly simplify and control the transport, heterogeneous crystallization and fluid dynamics of reactants, thereby fabricating compatible and position-adjustable MOF membranes in a time-saving and efficient manner. The thickness of ZIF-8 membranes can be controlled easily by adjusting sol concentration and coating time. The ZIF-8 membranes thus prepared show high permeances and selectivities in both single gas permeation and binary gas mixture separation, with selectivities of 3,126 and 73.4 for $H_2/C_3H_8$ and $C_3H_6/C_3H_8$ mixtures, respectively. In addition, GVD can in situ produce MOF membrane module with 30 polymeric hollow fibres and effective area of 340 cm$^2$ in scalability. The membrane

module shows hardly any reduction in selectivities compared with the small-scale membranes. Overall, the method described herein provides an alternative for the scalable and controllable production of ultrathin gas separation membranes with unique molecular sieving properties.

## Methods

**Zn-based gel fabrication**. For fabrication of the Zn-based sol, zinc acetate (5.0 g) was dispersed in ethanol (15 ml) and stirred at 60 °C for 0.5 h to obtain homogeneous white suspension[34]. Ethanolamine (1.5 ml) was dropwise added to the prepared suspension. Zinc acetate dihydrate was dissolved to produce transparent Zn-based sol. The Zn-based sol was stirred sequentially at 60 °C for 0.5 h for ageing. This sol concentration was labelled as one concentration unit (U). To achieve the Zn-based gel, the Zn-based sol was poured into a Petri dish and heat treated at 150 °C for drying. After heat treatment, the solvent was evaporated and the viscous white Zn-based gel was formed.

**Bulk ZIF-8 synthesis**. The Zn-based gel was placed in a multihole saucer. The holes of saucer benefit the diffusion of ligand vapour. Then, the Zn-based gel and saucer were sealed in a Teflon-lined stainless steel autoclave, which has been aforehand filled with some 2-methylimidazole. The Zn-based gel and pure ligand powder were contactless. The weight of 2-methylimidazole was larger than that of Zn-based gel to ensure complete reaction. The autoclave was heat treated at 150 °C for 2 h for crystallization. After reaction, the autoclave was cooled naturally to room temperature and the bulk ZIF-8 was obtained. The bulk ZIF-8 material was ground into powder. Ultimately, the ZIF-8 powder was washed by methanol for several times and dried for characterization.

**PVDF hollow fibre preparation and ammonition**. PVDF hollow fibres were prepared by typical wet-spinning process. The PVDF (16 wt%) and poly(ethylene glycol) (PEG; 3 wt%) was dissolved into N,N-dimethylacetamide (DMAc). The mixture was stirred at 60 °C to form uniform polymer solution. The solution was kept without stirring overnight to degas. Then, the degassed solution was extruded through a spinneret and passed through an air gap (15 cm) and into a water quench. The inner and outer coagulant solvents were both water with temperature of 50 °C. After coagulation, the membranes were washed with a large amount of water to remove the residual solvent and immersed into the water until use. The inner and outer diameters of the membrane were 0.8 and 1.5 mm, respectively. The porosity was ~75%. The molecular weight cutoff was ~150,000 Da. To improve the stability of substrate, PVDF hollow fibre was ammoniated by ethanediamine. In typical process, PVDF hollow fibres were cut into 4–5 cm and washed by water under ultrasonic treatment to remove the impurities. After natural drying, hollow fibres were immersed in ethanediamine solution (100 ml, 25%, v v$^{-1}$) in Teflon-lined stainless steel autoclave, and heat-treated at 150 °C for 20 h for ammonition. After reaction, the autoclave was cooled naturally and the hollow fibres were taken out. The white hollow fibres were changed to black brown. To remove unreacted ethanediamine, the ammoniated hollow fibres were washed by ethanol and water for three times, respectively. The purified hollow fibres were dried at room temperature for use.

**Zn-based gel coating**. After ammonition, the PVDF hollow fibre was immersed in the prepared Zn-based sol with various concentrations (from 0.1 to 2 U) and maintained for a certain time (from 2 to 30 s). Both two cross-sectional ends of the PVDF hollow fibre were sealed by a self-made Teflon stopper to prevent the formation of Zn-based gel on the inner surface. After immersion, the PVDF hollow fibre was taken out and purged by nitrogen to remove the redundant sol. The coated PVDF hollow fibre with sol was heat treated at 150 °C for 2 h to evaporate the solvent and obtain continuous Zn-based gel layer. In order to promote the uniform heat treatment of whole hollow fibres, a stainless steel wire with diameter of 0.5 mm was punctured through the tube side carefully and carried by a shelf. The thickness of the gel layers and ZIF-8 membranes was controlled by the sol concentration and the coating time. For synthesis of the MOF layer on inner surface of the hollow fibre, the sol was injected into the tube side of the substrate, then subjected to heat treatment for formation of gel.

**ZIF-8 membrane synthesis**. The 2-methylimidazole powder was put in a Teflon-lined stainless steel autoclave. The PVDF hollow fibre with Zn-based gel layer was placed vertically in a multihole saucer using a self-made Teflon holder. Then, the hollow fibre and saucer were sealed in the autoclave above the 2-methylimidazole powder. The Zn-based gel and 2-methylimidazole powder were contactless. The autoclave was heat treated at 150 °C for vapour deposition. The ZIF-8 membrane fabrication was executed with deposition time from 30 min to 6 h. After natural cooling, the gel was transformed to ZIF-8, and the MOF membrane was obtained. The 2-methylimidazole was reusable. Ultimately, the ZIF-8 membrane was washed by methanol for several times and dried at room temperature.

**Hollow fibre membrane module design**. PSF hollow fibres were cut into 25 cm for installation of membrane module, and washed by water under ultrasonic treatment to remove the impurities. After drying, 30 hollow fibres with an outside diameter of 0.16 cm were installed in a glass shell that possesses two nozzles, and glued together with epoxy resins as the module design basis. In order to prevent the blocking of epoxy resins in the tube side and improve the dispersion of hollow fibres in the whole glass shell, two ends of hollow fibres were tied tightly by rope. After curing, the redundant epoxy resins and the knots were sliced off. This process was carried out carefully to ensure the integrity of tube side. The glass shell has an inside diameter of 1.40 cm and length of 21 cm. The module has an effective mass-transfer length of 19 cm. The void fraction ($\varepsilon$), area per volume ($a$) and packing factor of the membrane module are 61%, 980 $m^2 \, m^{-3}$ and 2,455 $a\varepsilon^{-3}$, respectively.

**In situ Zn-based gel coating**. For in situ Zn-based gel coating, Zn-based sol (12 ml) was poured into the shell side of the membrane module. The module was vigorously shocked for 20 s to improve the evenly coating of hollow fibres. After coating, the sol was removed quickly and purged by nitrogen. In order to get rid of excess gel as quickly as possible, the module was placed with two nozzles in vertically downward direction during nitrogen purging process. Then, the coated membrane module with sol was heat treated at 150 °C for 2 h to evaporate the solvent and obtain continuous Zn-based gel layer. In this process, two nozzles of the glass shell were upward to promote solvent evaporation. After heat treatment, the colour of hollow fibres was changed from white to brown. This result should be explained by the reaction between the polymer and the ethanolamine that existed in Zn-based sol.

**ZIF-8 membrane module production**. The production of ZIF-8 membranes in module is described as follows. The 2-methylimidazole powder was put in two ligand containers. The ligand containers were hermetically connected with the two nozzles of the glass shell. Subsequently, the apparatus was heated for crystallization. In heat treatment, the ligand vapour was produced and flowed into the shell side of module, and then interacted with the Zn-based gel to form ZIF-8. In vapour deposition, two ligand containers were placed downside to promote vapour diffusion and prevent the flow of melting ligand in shell side. After reaction for 2 h, the module was cooled naturally. Ultimately, the membrane module was washed for several times and dried at room temperature.

**Gas permeation measurements**. The transport behaviours of various gases through the ZIF-8 membrane were investigated by constant-pressure, variable-volume method. For single ZIF-8 hollow fibre membrane, it was put in a permeation module and sealed by epoxy resins. For ZIF-8 hollow fibre membrane module, it was employed as permeation module directly. Before measurement, the permeation module was rinsed by measured gas. The feed gas was fed to the shell side, and the permeate gas was collected at the tube side of the hollow fibre membrane. The effective area was calculated by the outer surface of the membrane. Gas flow rates were measured by a bubble flow metre. The gas permeation experiment was carried out with different kinetic diameters (KD) of gases in the following order: $H_2$ (KD: 0.289 nm), $CO_2$ (KD: 0.33 nm), $O_2$ (KD: 0.346 nm), $N_2$ (KD: 0.364 nm), $CH_4$ (KD: 0.38 nm), $C_3H_6$ (KD: 0.40 nm), $C_3H_8$ (KD: 0.43 nm), $C_3H_8$, $C_3H_6$, $CH_4$, $N_2$, $O_2$, $CO_2$ and $H_2$. The data were read and recorded until the system running stably. The gas permeation data were calculated by averaging the measured values of two cycles. To obtain the upper bound of polymeric membranes, the membrane thickness of the polymeric membrane is assumed as 1 μm. For investigation of the stability of ZIF-8 membrane, permeation data were recorded every 2 h. After recording for four times, another gas was fed into the shell side and the permeation data were read. To keep the system running stably, the permeation data in first 2 h were not recorded after changing the feed gas. This experiment with two cycles of gas permeation lasted for 274 h.

**Characterizations**. The porosity of the prepared ZIF-8 was investigated by $N_2$ adsorption–desorption isotherms, performed on a Micromeritics-Accelerated Surface Area and Porosimetry system (ASAP 2020M+C, Micromeritics Instrument Co., USA). Measurements were carried out at 77 K held using a liquid nitrogen bath. Before the analysis, the samples were degassed in vacuum at 150 °C for 12 h. BET method was used to calculate the specific surface areas in the $P/P_0$ range of 0.05–0.1. The crystalline structure of the as-synthesized materials was studied by powder XRD. XRD data were recorded by a PNAlytical X' Pert PRO X-ray diffractometer with CuKα radiation (λ = 0.154056 nm) at 40 kV and 40 mA. The data were recorded from 5° to 40° in a continuous scanning mode. Scan speed and step size was set as 1 s per step and 0.02°, respectively. For characterization of the hollow fibre membranes, the membranes were ground in liquid nitrogen to prepare the powder samples. The change of the chemical structure was tested by FTIR. A Nicolet 6700 FTIR–Attenuated Total Reflectance (Thermo Scientific Co.) spectrophotometer was employed to record infrared spectra. The samples were compacted into discs by a powder tablet press for measurement. The morphology of the membrane was observed by SEM. The SEM (S-4700, Hitachi, Japan) was employed to observe the morphologies of the membranes. Accelerating voltage was set as 15 kV. All the prepared samples were coated with an ultrathin layer of platinum using an ion sputter coater (E-1045, Hitachi) to minimize charging

effects. In order to keep the cross-sectional morphology of the hollow fibre membrane, it was freeze-fractured in liquid nitrogen. Surface morphologies of the prepared membranes were characterized by using an AFM (Bioscope Catalyst Nanoscope-V, Bruker, USA). The samples were fixed on glass slides for measurement. Root mean square roughness ($R_q$) and arithmetic average roughness ($R_a$) were calculated using the NanoScope Analysis software.

**Data availability**. The data that support the findings of this study are available within the article (and Supplementary Information Files) or available from the corresponding author on reasonable request.

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

## Acknowledgements

This work was financially supported by Jinan University (Grant No. 88016674) and the National Natural Science Foundation of China (Grant Nos 21236008 and 21476206).

## Author contributions

W.L. and G.Z. conceived the research idea and formulated the project. W.L., P.S., Z.L., Z.X. and J.Z. performed experiments, including gel fabrication, ZIF-8 synthesis, gel coating, module design and membrane deposition. W.L., Z.L., F.W., H.O. and G.Z. carried out characterizations and analyses including gas adsorption, scanning electron microscopy, powder diffraction, Fourier transform infrared spectroscopy, atomic force microscopy and gas permeation data. W.L., G.Z. and E.Z. wrote the paper. All authors contributed to the revising the paper.

## Additional information

**Competing interests:** The authors declare no competing financial interests.

