## [Peer Review File · Nature Communications]

Reviewers' comments:

Reviewer #1 (Remarks to the Author):

This is a nice membrane fabrication and characterization work involving the production of ZIF membranes with smaller thickness and higher permeance. I feel this work fits in a more specialist applied journal and is not suited for a Nature X journal. None of the concepts or issues are new: vapor, liquid, and solid phase transformation of ZnO and other Zn compounds to ZIFs is already known. Use of ZnO and other Zn-rich coatings/layers to seed ZIF-8 membrane growth has been shown by several authors. Use of hollow fibers to grow ZIF-8 membranes is also known. The fibers used here seem to be standard commercial fibers (see specific comment #1 below) and there seems no novelty in engineering these fiber supports.

In terms of results, the demonstration of higher flux (due to lower membrane thickness) is interesting, and also the demonstration of the hollow fiber module. There is an element of engineering/scale-up advance in this work, but this fits better in an applied/engineering journal. No fundamental issues/advances were identified. The manuscript also has some serious issues in terms of the data and discussion, as listed below:

- No information could be found regarding the hollow fiber supports. Were they commercially obtained, or produced by the authors? If the latter, what are the details of the spinning process? What are the characteristics of these fibers (porosity, pore size, permeance, etc)? Were the supports engineered in some way, or are they 'generic' fiber materials ?
- It is hard to understand why there is no separation data reported in the manuscript. The authors report single-component measurements on different gases. I do not think this is sufficient for the claim that the membranes are good for separation.
- Similarly, how stable are the membranes at realistic conditions such as higher pressure ? The measurements seem to be done at 1 bar pressure, which is not really relevant. E.g., for propylene/propane separation one needs feed pressures in the 6-9 bar range. As reported by several authors, the ZIF-8 membrane performance sharply decreases with increasing pressure... a selectivity of ~ 70 at 1 bar could turn into < 5 at 9 bar.
- There are no error bars on any of the data. It is unclear how reproducible or reliable the data are. Moreover, nobody else can reproduce it since the authors do not mention any details of the fiber supports.

In summary I do not think there is a sufficient conceptual advance over the existing literature to justify publication in Nature X journals. The paper could fit nicely (after considerable revision and addition of important data) in a more applied/specialist journal.

Reviewer #2 (Remarks to the Author):

It is indeed an important paper which report a remarkable progress in the preparation of thin and perfect high-flux membranes with good selectivity.

There are current efforts to prepare < 100 nm MOF membrane layers, the most important ones are correctly given in the manuscript. However, I miss the SURMOFs made by layer-by-layer technique.

A great advantage of the GVD method is the coating of organic fibers. Is also inside-coating possible? Can the layer be annealed if scratched or otherwise mechanically damaged? Can authors say something on long-time stability? Steam stability of ZIF-8 was and will be an issue.

Reviewer #3 (Remarks to the Author):

The paper is well written and new results are presented related to a very efficient methodology MOF membrane preparation. Of particular importance is the scale up of the preparation procedures with up to 30 polymeric HFs. The paper can be published in NC after next suggested revision:

1. Abstract. I can follow this "...selectivity of as high as 3400, 1030 and 70 toward H₂, CO₂ and C₃H₆ over C₃H₈, respectively".
2. Introduction. When talking about HFs, mention about MOF continuous hollow fibers by direct MOF crystallization: e.g. works by Brown et al., Cacho-Bailo et al., Biswal et al. And also those corresponding to MMMs as HFs (e.g. Oguz et al.)
3. Describe panel k in the corresponding Figure 2 caption.
4. Alto, it is not clear what "0.1 U" is, since it appears for the first time in Fig.1 caption without any explanation (only given on page 7).
5. Fig. 3. For clarity out the permance units without multiplying by a factor. In fact this factor changes from one plot to another (10⁻⁸, 10⁻⁷).
6. Check Fig. 3 caption: there is neither literature comparison nor upper-bound in Fig. 3c!
7. Fig. 3. Say clearly in the caption whether this is or no single o mixture gas permeation.
8. H₂/CH₄ (17.9-30.7) and CO₂/CH₄ (6.3-9.4) selectivities are far below the present state of the art regarding ZIF hollow fiber membranes: in particular, recent works with ZIF-8 and ZIF-93, not mentioned here, showed values close ca. 100 and 20, respectively.

Response to the comments of reviewers on
"Ultrathin metal-organic framework membrane production by gel-vapour deposition"
(NCOMMS-17-01402-T) by Li, et al.

Reviewer #1:

This is a nice membrane fabrication and characterization work involving the production of ZIF membranes with smaller thickness and higher permeance. I feel this work fits in a more specialist applied journal and is not suited for a Nature X journal. None of the concepts or issues are new: vapor, liquid, and solid phase transformation of ZnO and other Zn compounds to ZIFs is already known. Use of ZnO and other Zn-rich coatings/layers to seed ZIF-8 membrane growth has been shown by several authors. Use of hollow fibers to grow ZIF-8 membranes is also known. The fibers used here seem to be standard commercial fibers (see specific comment #1 below) and there seems no novelty in engineering these fiber supports.

In terms of results, the demonstration of higher flux (due to lower membrane thickness) is interesting, and also the demonstration of the hollow fiber module. There is an element of engineering/scale-up advance in this work, but this fits better in an applied/engineering journal. No fundamental issues/advances were identified. The manuscript also has some serious issues in terms of the data and discussion, as listed below:

Response:

We thank the reviewer for his/her comments. We will further illustrate the innovations of this work in the following statement.

First, "None of the concepts or issues are new: vapor, liquid, and solid phase transformation of ZnO and other Zn compounds to ZIFs is already known." The focus of this work is realizing the production of ultrathin MOF membranes rather than MOFs. The growth mechanism, requirements and target of MOF membranes are vastly different from that of MOFs. So far, preparation of MOF membranes with nanometer-thick layers or in a scalable route has remained a great challenge. We have fabricated the ultrathin ZIF-8 membranes, and demonstrated the in-situ scale-up processing

of ZIF-8 membrane module with 30 polymeric hollow fibers and membrane area of 340 square centimeters. Moreover, the prepared membranes exhibit excellent permeance and selectivity.

Second, “Use of ZnO and other Zn-rich coatings/layers to seed ZIF-8 membrane growth has been shown by several authors”. Although, as reviewers’ comments, Zhang et al., Li et al. and Yu et al. fabricated the ZnO as seeds to obtain ZIF-8 membranes (Zhang, X. *et al. Chem. Mater.* 26, 1975-1981 (2014); Li, W. *et al. Chem. Commun.* 50, 9711-9713 (2014); Yu, J. *et al. Chem. Eng. Sci.* 141, 119-124 (2016).), Caro group grew the ZnAl-CO₃ LDH as buffer layer to fabricate the ZIF-8/LDH composite membranes (Liu, Y. *et al. J. Am. Chem. Soc.* 36, 14353-14356 (2014); Liu, Y. *et al. Angew. Chem. Int. Ed.* 54, 3028-3032 (2015).). As shown in Response Table 1, in those studies, the ZnO and other Zn-rich coatings/layers are usually only served as seed layer. For obtaining the continuous membranes, second hydro/solvothermal treatment is necessary. As well as, the Zn-rich layers are often prepared by the complex synthesis procedures such as hydro/solvothermal treatment, sputtering deposition and calcination, and usually should be subjected to activation before MOF membrane growth. GVD is transformation of the Zn-based gel to the MOF membrane directly through ligand vapor deposition. GVD is performed by scalable sol-gel coating and solvent-free vapor deposition, avoids the formation of cracks, reduces the consumption of expensive MOF precursors, shorten the synthesis time and can manipulate the position of MOF layers. Moreover, compared with the Zn-rich coatings/layers to seed ZIF-8 membrane growth, which usually prepared the micrometer-thick membrane in bench scale with membrane area of several square centimeters, GVD can produce the nanometer-thick membranes and shows the great potential in industrial applications.

Third, “use of hollow fibers to grow ZIF-8 membranes is also known.” As mentioned above, the key find of this work is production of ultrathin MOF membranes by GVD. The fiber substrates have been employed to show the utility and distinctive features of GVD, because of their benefits of low cost, large membrane area per volume and commendable processing ability. We can also synthesize the ZIF-8 membranes on other substrates. For examples, we have fabricated the ZIF-8 membrane on AAO substrates (see explanation below).

Response Table 1 | Synthesis procedures and properties of the ZIF-8 membranes prepared by ZnO and other Zn-rich coating/layer seeding in literatures.

Seed/Method	Activation/Synthesis methods	Thickness	Area	Ref
ZnO/Slip-casting and hydrothermal	Solvothermal/Solvothermal	6 μm	5.2 cm^2	1
ZnO/Slip-casting and calcination	Solvothermal/Solvothermal	8 μm	5.6 cm^2	2
ZnO/Solvothermal	-/Solvothermal	50 μm	1.8 cm^2	3
ZnO/Sputtering	Hydrothermal/Hydrothermal	2.5 μm	3.8 cm^2	4
LDH/Hydrothermal	-/Solvothermal	20 μm +1 μm (LDH)	2.5 cm^2	5
LDH/Hydrothermal	-/Solvothermal	1.1 μm +1.3 μm (LDH)	2.5 cm^2	6

References

1. Zhang, X. F. *et al.* New membrane architecture with high performance: ZIF-8 membrane supported on vertically aligned ZnO nanorods for gas permeation and separation. *Chem. Mater.* **26**, 1975-1981 (2014).
2. Zhang, X. F. *et al.* A simple and scalable method for preparing low-defect ZIF-8 tubular membranes. *J. Mater. Chem. A* **1**, 10635-10638 (2013).
3. Li, W. *et al.* Non-activation ZnO array as a buffering layer to fabricate strongly adhesive metal-organic framework/PVDF hollow fiber membranes. *Chem. Commun.* **50**, 9711-9713 (2014).
4. Yu, J., Pan, Y., Wang, C. & Lai, Z. ZIF-8 membranes with improved reproducibility fabricated from sputter-coated ZnO/alumina supports. *Chem. Eng. Sci.* **141**, 119-124 (2016).
5. Liu, Y., Wang, N. Y., Pan, J. H., Steinbach, F. & Caro, J. In-situ synthesis of MOF membranes on ZnAl-CO₃ LDH buffer layer-modified substrates. *J. Am. Chem. Soc.* **136**, 14353-14356 (2014).
6. Liu, Y. *et al.* Remarkably enhanced gas separation by partial self-conversion of a laminated membrane to metal-organic frameworks. *Angew. Chem. Int. Ed.* **54**, 3028-3032 (2015).

-No information could be found regarding the hollow fiber supports. Were they commercially obtained, or produced by the authors? If the latter, what are the details of the spinning process? What are the characteristics of these fibers (porosity, pore size, permeance, etc)? Were the supports engineered in some way, or are they 'generic' fiber materials?

Response:

PVDF hollow fiber supports were prepared by a typical wet-spinning process. The PVDF 16 wt% and PEG 3 wt% was dissolved into DMAc. The mixture was stirred at 60 °C to form uniform polymer solution. The solution was kept without stirring for overnight to degas. Then the degassed solution was extruded with pressure of 2 bar through a spinneret and passed through an air gap (15 cm) and into a water quench. The inner and outer coagulant solvents were both water with temperature of 50 °C. After coagulation, the hollow fibers were washed with a large amount of water to remove the residual solvent and immersed into the water until use. The inner and outer diameters of the membrane were 0.8 and 1.5 mm, respectively. The porosity was about 75%. The molecular weight cut off was about 150000 Da.

For improving the stability of the PVDF hollow fiber, it was ammoniated by ethanediamine solution. PVDF hollow fibers were cut into 4-5 cm and washed by water under ultrasonic treatment to remove the impurities. After natural drying, 20 hollow fibers were immersed in 100 ml ethanediamine solution (25 %, v/v) in Teflon lined stainless steel autoclave, and heat-treated at 150 °C for 20 h for ammoniation. After reaction, the autoclave was cooled naturally and the hollow fibers were taken out. The white hollow fibers were changed to black brown. To remove unreacted ethanediamine, the ammoniated hollow fibers were washed by ethanol and water for three times, respectively. After ammoniation, the ethanediamine displaced the fluorine atom and made the PVDF molecules cross-link together, the hydrofluoric acid of PVDF was removed to form a -C=C- bond when presented in alkaline. Thus the stability of the ammoniated PVDF can be greatly enhanced. Moreover, after ammoniation, the dense skins of the PVDF hollow fiber of ammoniated membranes changed to the polymer nanoparticle layer, and the diameter falls in the range of 20 to 50 nm. The ammoniated PVDF hollow fiber had H₂ permeance of $9 \times 10^{-5} \text{ mol m}^{-2} \text{ s}^{-1} \text{ Pa}^{-1}$ and H₂/C₃H₈ selectivity of 3.1.

We have revised the manuscript.

-It is hard to understand why there is no separation data reported in the manuscript. The authors report single-component measurements on different gases. I do not think this is sufficient for the claim that the membranes are good for separation.

Response:

We have fabricated additional ZIF-8 membrane to separate the binary propylene/propane mixture as reviewers' suggestion. In single-component measurement, the membrane shows H₂ and C₃H₆ permeances of $126.7 \times 10^{-7} \text{ mol m}^{-2} \text{ s}^{-1} \text{ Pa}^{-1}$ and $2.9 \times 10^{-7} \text{ mol m}^{-2} \text{ s}^{-1} \text{ Pa}^{-1}$, as well as H₂/C₃H₈ and C₃H₆/C₃H₈ selectivities of 3011 and 68.9, respectively. In binary mixture separation, the similar permeances ($117.8 \times 10^{-7} \text{ mol m}^{-2} \text{ s}^{-1} \text{ Pa}^{-1}$ for H₂ in H₂/C₃H₈ separation and $2.8 \times 10^{-7} \text{ mol m}^{-2} \text{ s}^{-1} \text{ Pa}^{-1}$ for C₃H₆ in C₃H₆/C₃H₈ separation) and selectivities (3126 for H₂/C₃H₈ and 73.4 for C₃H₆/C₃H₈) are achieved (Supplementary Fig. 10, in revised manuscript). These results are sufficient for demonstrating that the membranes are good for separation. We have revised the manuscript.

-Similarly, how stable are the membranes at realistic conditions such as higher pressure ? The measurements seem to be done at 1 bar pressure, which is not really relevant. E.g., for propylene/propane separation one needs feed pressures in the 6-9 bar range. As reported by several authors, the ZIF-8 membrane performance sharply decreases with increasing pressure... a selectivity of ~70 at 1 bar could turn into < 5 at 9 bar.

Response:

We have investigated the effect of feed pressure on C₃H₆ permeance and C₃H₆/C₃H₈ selectivity of ZIF-8 membrane (Supplementary Fig. 11, in revised manuscript). With the increase of feed pressure, the C₃H₆/C₃H₈ selectivity and the C₃H₆ permeance both decrease, which is similar to the observation in previous studies (Liu, D. *et al. J. Membr. Sci.* 451, 85-93 (2014); Yu, J. *et al. Chem. Eng. Sci.* 141, 119-124 (2016).). This phenomenon is attributed to the gate opening of window of ZIF-8 structure and competitively diffusion of the C₃H₆ and C₃H₈ through

the membrane. However, the prepared ZIF-8 membrane still shows good C_3H_6/C_3H_8 selectivity of 35.8 and C_3H_6 permeance of $2.3 \times 10^{-7} \text{ mol m}^{-2} \text{ s}^{-1} \text{ Pa}^{-1}$ at 6 bar.

Supplementary Figure 11 | Effect of feed pressure on C_3H_6 permeance and C_3H_6/C_3H_8 mixture selectivity of ZIF-8 membrane. The membrane was the additional ZIF-8 membrane 3, as shown in Supplementary Figure 10.

- There are no error bars on any of the data. It is unclear how reproducible or reliable the data are. Moreover, nobody else can reproduce it since the authors do not mention any details of the fiber supports.

Response:

To further study the reproducibility of the ZIF-8 membranes, additional three membranes have been prepared. The gas permeation results are presented in Supplementary Figure 10 (in revised manuscript). The H_2 permeances of these three membranes are 106.5, 122.9 and $126.7 \times 10^{-7} \text{ mol m}^{-2} \text{ s}^{-1} \text{ Pa}^{-1}$, and the corresponding selectivities of $H_2/C_3H_8-C_3H_6/C_3H_8$ are 2701-67.2, 2845-64.4, and 3012-68.9, respectively. These results demonstrate the good reproducibility of the membranes.

As for membrane synthesis, the concentration of the sol and the pore size of the substrates are

critical for obtaining the continuous ZIF-8 membranes. As described in manuscript, when the Zn-based sol concentration is diluted to 0.05 U (0.05/3 g ml⁻¹ of zinc acetate dihydrate/ethanol), some pinholes are formed in the membrane. This can be attributed to the high porosity of substrates and low viscosity of sol. The surface of ammoniated PVDF substrate is composed of nanoparticles with diameter of 20-50 nm. Low viscosity lead to small loading and fast spread of Zn-based sol in porous PVDF substrates (Supplementary Fig. 7 and Response Fig. 2a-d). Besides adjustment of the sol concentration, we have further coated the sol with concentration of 1 U (1/3 g ml⁻¹ of zinc acetate dihydrate/ethanol) on substrates with different pore sizes. As shown in Response Figure 2e-h, when the sol is coated on the substrate with 180 nm pores, no continuous membrane is obtained. However, when the substrate with 25 nm pores is employed, the formed ZIF-8 membrane is continuous.

Supplementary Figure 11 | Gas permeation behaviours of additional three ZIF-8 membranes.

These ZIF-8 membranes were prepared with sol concentration of 1 U, coating time of 2 s and deposition time of 2 h.

Response Figure 2 | **a,c**, Schematic diagram and **b,d**, top view SEM image of **a,b**, the noncontinuous ZIF-8 membrane fabricated with sol concentration of 0.05 U and **c,d**, the continuous ZIF-8 membrane fabricated with sol concentration of 0.1 U. **e,g**, Schematic diagram and **f,h**, top view SEM image of **e,f**, the noncontinuous ZIF-8 membrane on substrates with pore size of 180 nm and **g,h**, the continuous ZIF-8 membrane on substrates with pore size of 25 nm; the sol concentration is 1 U.

These ZIF-8 membranes were prepared with coating time of 2 s and deposition time of 2 h. SEM images of the ZIF-8 membrane on AAO substrates.

In summary I do not think there is a sufficient conceptual advance over the existing literature to justify publication in Nature X journals. The paper could fit nicely (after considerable revision and addition of important data) in a more applied/specialist journal.

Reviewer #2:

It is indeed an important paper which report a remarkable progress in the preparation of thin and perfect high-flux membranes with good selectivity.

There are current efforts to prepare < 100 nm MOF membrane layers, the most important ones are correctly given in the manuscript. However, I miss the SURMOFs made by

layer-by-layer technique.

Response:

We thank the reviewer for his/her positive comments. Layer-by-layer (LBL) synthesis is a great strategy to obtain the SURMOFs. LBL synthesis is mild, simple and controllable. Because of the inconsecutive crystallization process, the thickness of the prepared MOF layer can be precisely controlled by the growth cycles. However, it is usually applied to obtain the ultrathin MOF films for electronic, optical and sensing applications (Shekhah, O. *et al. Nature Mater.* **8**, 481-484 (2009); Talin, A. A. *et al. Science* **343**, 66-69 (2014); Sakaida, S. *et al. Nature Chem.* **8**, 377-383 (2016).); Zhuang, J. L. *et al. Coordin. Chem. Rev.* **307**, 391-424 (2016).) rather than MOF membranes for separation. Although few papers have reported the fabrication of MOF separation membranes by LBL synthesis, the prepared MOF membrane have not showed the separation performance as expected and the thickness of the MOF layer are also not ultrathin (< 100 nm). Shekhah *et al.* prepared a thin and continuous ZIF-8 layer on alumina substrate by LBL (Shekhah, O. *et al. Chem. Commun.* **50**, 2089-2092 (2014).). After growth for 150-300 cycles, the ZIF-8 membranes with thickness of 0.5-1.6 μm were achieved. The membrane showed a small selectivities (H_2/CO_2 -5, H_2/N_2 -11, H_2/CH_4 -12, $\text{H}_2/\text{C}_3\text{H}_8$ -70 and $\text{C}_3\text{H}_6/\text{C}_3\text{H}_8$ -3.5) and low gas permeances ($1.9 \times 10^{-8} \text{ mol m}^{-2} \text{ s}^{-1} \text{ Pa}^{-1}$ for H_2 and $0.06 \times 10^{-8} \text{ mol m}^{-2} \text{ s}^{-1} \text{ Pa}^{-1}$ for C_3H_6) The authors attributed these phenomena to the difference in internal structure of the prepared ZIF-8 membrane. Nagaraju *et al.* also employed the LBL method to fabricated MOF membranes (Nagaraju, D. *et al. J. Mater. Chem. A* **1**, 8828-8835 (2013).). After synthesis, a CuBTC membrane with a thickness of 7.2 μm was prepared. The membrane showed a H_2 permeance of $7.9 \times 10^{-8} \text{ mol m}^{-2} \text{ s}^{-1} \text{ Pa}^{-1}$, and $\text{H}_2/\text{C}_3\text{H}_6$ selectivity of 5.7, respectively. Compared with the CuBTC membranes prepared by other methods, the selectivity is moderate and the permeance is low.

The related parts in the manuscript have been revised.

A great advantage of the GVD method is the coating of organic fibers. Is also inside-coating possible? Can the layer be annealed if scratched or otherwise mechanically damaged? Can authors say something on long-time stability? Steam stability of ZIF-8 was and will be an issue.

Response:

Outer-surface membranes provide larger effective area than inner-surface membranes, and inner-surface membranes are likely to show greater mechanical stability. Benefitted from the simple coating process, the position of MOF layers is manipulated at the inner and outer surfaces of the hollow fibers. In fact, we have also fabricated the ZIF-8 layer on inner surfaces. SEM images of the inner surface and ZIF-8 layer on inner surface of PVDF hollow fiber are shown in Supplementary Figure 1d,e. After membrane fabrication, a continuous ZIF-8 membrane is synthesized on inside of the PVDF membranes. The XRD pattern also demonstrates the transformation of the gel to ZIF-8 (Supplementary Fig. 1f).

As like the MOF membranes and zeolite membranes reported in previous literatures, ZIF-8 membranes may not be annealed after scratched or otherwise mechanically damaged, because the ZIF-8 is a kind of relatively stiff crystal. However, because some ZIF-8 crystals are injected into the hollow fiber, which can greatly enhance the adhesion between the MOF layers and substrates, the prepared membranes have excellent mechanical stability. In fact, before the gas permeation measurements, the ZIF-8 membranes have been washed by methanol for several times, the first washing process are carried out under the ultrasonic condition for the 5-6 min. The good separation performance of the ultrasonically treated ZIF-8 membranes illustrates their excellent mechanical stability.

We have characterized the permeances of various gases through ultrathin ZIF-8 membranes for two measurement cycles of 274 h (Fig. 3a). Gas permeances show hardly any variation after measurement for 274 h. This result displays the extraordinary long-time stability of ZIF-8 membrane. The related parts in the manuscript have been revised.

Supplementary Figure 1 | Top view SEM image of **d**, the inner surface and **e**, the ZIF-8 layer on inner surface of PVDF hollow fiber. **f**, XRD patterns of the simulated ZIF-8 and the ZIF-8 membrane on inner surface of PVDF hollow fiber.

Reviewer #3 (Remarks to the Author):

The paper is well written and new results are presented related to a very efficient methodology MOF membrane preparation. Of particular importance is the scale up of the preparation procedures with up to 30 polymeric HFs. The paper can be published in NC after next suggested revision:

1. **Abstract.** I can follow this "...selectivity of as high as 3400, 1030 and 70 toward H₂, CO₂ and C₃H₆ over C₃H₈, respectively".

Response:

We thank the reviewer for his/her positive comments. The "...selectivity of as high as 3400, 1030 and 70 toward H₂, CO₂ and C₃H₆ over C₃H₈, respectively" have been revised to "...H₂/C₃H₈, CO₂/C₃H₈ and C₃H₆/C₃H₈ selectivities of as high as 3400, 1030 and 70, respectively"

2. **Introduction.** When talking about HFs, mention about MOF continuous hollow fibers by direct MOF crystallization: e.g. works by Brown et al., Cacho-Bailo et al., Biswal et al. And also those corresponding to MMMs as HFs (e.g. Oguz et al.)

Response:

We have cited the mentioned works in related parts of the revised manuscript. The related

parts have been revised.

19. Biswal, B. P., Bhaskar, A., Banerjee R. & Kharul, U. K. Selective interfacial synthesis of metal–organic frameworks on a polybenzimidazole hollow fiber membrane for gas separation. *Nanoscale* **7**, 7291-7298 (2015).
20. Brown, A. J. *et al.* Continuous polycrystalline zeolitic imidazolate framework-90 membranes on polymeric hollow fibers. *Angew. Chem. Int. Ed.* **51**, 10615-10618 (2012).
21. Cacho-Bailo, F. *et al.* High selectivity ZIF-93 hollow fiber membranes for gas separation. *Chem. Commun.* **51**, 11283-11285 (2015).
22. Cacho-Bailo, F. *et al.* On the molecular mechanisms for the H₂/CO₂ separation performance of zeolite imidazolate framework two-layered membranes. *Chem. Sci.* **8**, 325-333 (2017).
24. Dai, Y., Johnson, J. R., Karvan, O., Sholl, D. S. & Koros, W. J. Ultem[®]/ZIF-8 mixed matrix hollow fiber membranes for CO₂/N₂ separations. *J. Membr. Sci.* **401-402**, 76-82 (2012).

3. Describe panel k in the corresponding Figure 2 caption.

Response:

We have revised the related parts. Figure 2k, Grain size distribution of the Zn-based gel layer and the ZIF-8 membrane from AFM data.

4. Alto, it is not clear what “0.1 U” is, since it appears for the first time in Fig.1 caption without any explanation (only given on page 7).

Response:

We have revised the manuscript. U is employed to describe the concentration of the sol. For convenience of discussions, 1/3 g ml⁻¹ of zinc acetate dihydrate/ethanol sol is defined as one concentration unit (U) in the this manuscript. 0.1 U means that the zinc acetate dihydrate/ethanol is 0.1/3 g ml⁻¹.

5. Fig. 3. For clarity out the permance units without multiplying by a factor. In fact this factor changes from one plot to another (10⁻⁸, 10⁻⁷).

Response:

We have revised the manuscript as shown in Figure 3, Supplementary Figure 9, Supplementary Figure 12 and Supplementary Table 1-3.

6. Check Fig. 3 caption: there is neither literature comparison nor upper-bound in Fig. 3c!

Response:

We have revised the manuscript. As shown in Figure 3, “**b**, Comparison of ZIF-8 membranes here with polymeric³⁷, carbon³⁶ and other MOF membranes in literatures for C₃H₆/C₃H₈ system. The black line is the upper-bound of polymeric membranes³⁷. The numerical data is shown in Supplementary Table 3. The permeance is calculated from the permeability by assuming membrane thickness of 1 μm. **c**, Gas permeation behaviour of different gas through ZIF-8 membranes prepared with various sol concentrations of 0.1 U, 1 U, 1.5 U and 2 U. These ZIF-8 membranes were prepared with coating time of 2 s and vapour deposition time of 2 h.”

7. Fig. 3. Say clearly in the caption whether this is or no single or mixture gas permeation.

Response:

We have revised the caption of Figure 3.

8. H₂/CH₄ (17.9-30.7) and CO₂/CH₄ (6.3-9.4) selectivities are far below the present state of the art regarding ZIF hollow fiber membranes: in particular, recent works with ZIF-8 and ZIF-93, not mentioned here, showed values close ca. 100 and 20, respectively.

Response:

We have revised the manuscript. The permeation data of these two membranes and some related membranes are list in Supplementary Table 1 and Supplementary Table 2.

Reviewers' comments:

Reviewer #1 (Remarks to the Author):

The authors have responded well to the reviewer comments. This paper is now much more technically credible and could be valuable for ZIF-8 gas separation membrane developments.

I still believe this paper fits better in an applied journal considering the extent of the previous literature (papers and patents) which already cover all the main ideas - but I leave that decision to the editors.

Scientifically, I think a few revisions/clarifications are still required:

1) The authors have now described the detailed method for the PVDF fiber production, which is good. However, this method (spinning followed by ammoniation) seems to be the same as that already published in previous works (I don't know if it is the same authors or someone else). The previous works should be cited in this discussion, and if it the same method then that should be clearly stated.

2) It is good that the authors tested the membranes at higher pressure. They found that C₃H₆/C₃H₈ selectivity reduces to ~30 at 6 bar. My prediction is that the selectivity will be about 10 at 9-10 bar, which is the real pressure of interest. Such a membrane would not be very useful since many stages would be required to get high-purity propylene. Irrespective of high membrane flux, the interstage compression requirements would greatly exceed the cost of the membrane itself.

Anyhow, my revision request here is again related to citation. The authors correctly mention two previous papers that also found a drastic reduction in selectivity at higher pressure. However, they did not cite a key recent paper by K. Eum et al (ACS Applied Materials & Interfaces, 8 (38), 25337-25342, 2016) where selectivities of 90+ were maintained at 9 bar. This should be cited/discussed. Similar to Eum's paper, the present authors also find slight *reduction* in permeance at higher pressures - this is different from the other two papers in which the permeance increased. The explanations for the two types of behavior are different.

3) In the cases where the authors have provided additional permeation data (e.g., binary permeation, higher pressures), they should clarify whether these data were collected with the single-fiber membranes or with the 30-fiber modules. It seems the former, but this should be stated in the discussion.

Reviewer #2 (Remarks to the Author):

Authors did a proper review - accept

Reviewer #3 (Remarks to the Author):

I am partially satisfied with the answers given by the authors. It is OK regarding points 1-7. However, regarding point 8, these membranes even if prepared by a new method show low selectivities (e.g. in case of CO₂/CH₄ in the 6.3-9.5 range) as compared to those in the literature. In addition, this gas separation performance was obtained from single gas permeance measurements and not from mixture separation experiments, as far as I can understand.

Reviewer #1

I still believe this paper fits better in an applied journal considering the extent of the previous literature (papers and patents) which already cover all the main ideas - but I leave that decision to the editors.

Response/action: Preparation of defect-free MOF membranes with nanometer-thick layers or in a scalable route has remained a great challenge (Brown, A. J. *et al. Science* **345**, 72-75 (2014); Peng, Y. *et al. Science* **346**, 1356-1359 (2014); Hu, Y. X. *et al. Angew. Chem. Int. Ed.* **55**, 204-2052 (2016); Eum, K. *et al. Adv. Funct. Mater.* **26**, 5011-5018 (2016); Denny, M. S. *et al. Nature Rev. Mater.* **1**, 16078 (2016); Koros, W. J. *et al. Nat. Mater.* **16**, 289-297 (2017)). In the present study, we fabricated the thinnest ZIF-8 membranes, which exhibited excellent separation performance, and demonstrated the in-situ scale-up processing ability of GVD.

More importantly, no study has so far employed the sol-gel method or vapor deposition to fabricate MOF separation membranes. **This is the first report on synthesis of MOF separation membranes with gel as the sole metal source.** The sol-gel method is widely employed to fabricate coating layers, and can be controlled and enlarged easily. Gel deposition hardly relies on the chemical properties of substrates, the gel layers can be deposited on various substrates. In comparison with other Zn-rich coatings/layers in solid state, which can also be used to prepare MOF membranes as seeds after complex and harsh synthesis, the sol-gel method is much more straightforward. **This is also the first report on synthesis of MOF separation membranes by vapor deposition.** For vapor deposition, the complex reactant transport, heterogeneous crystallization and fluid dynamics can be greatly simplified. The process of GVD completely avoids the swelling of polymeric hollow fibers, thus preventing the formation of cracks in MOF membranes. The consumption of expensive MOF precursors can be greatly reduced because precursors are reusable and no bulk MOF grows. In fact, we also tried to synthesize MOF membranes by solvothermal-treating gel-coated hollow fibers, but no continuous membranes were formed as the gel in 2-methylimidazole solution was dismissed. These features illustrate the important contribution of this work to the field of membrane science, which should be helpful for scalable production of ultrathin MOF membranes with precise molecular sieving properties.

We have revised the manuscript.

1) The authors have now described the detailed method for the PVDF fiber production, which is good. However, this method (spinning followed by ammoniation) seems to be the same as that already published in previous works (I don't know if it is the same authors or someone else). The previous works should be cited in this discussion, and if it the same method then that should be clearly stated.

Response/action: Because the present study was focused on the development of GVD for synthesis of MOF membranes, we have not emphasized the detailed procedures of hollow fiber fabrication and modification in the manuscript, though both the detailed wet-spinning method and the ammoniation were proposed by ourselves. We have cited the previous studies in the revised manuscript.

2) It is good that the authors tested the membranes at higher pressure. They found that C_3H_6/C_3H_8 selectivity reduces to ~ 30 at 6 bar. My prediction is that the selectivity will be about 10 at 9-10 bar, which is the real pressure of interest. Such a membrane would not be very useful since many stages would be required to get high-purity propylene. Irrespective of high membrane flux, the interstage compression requirements would greatly exceed the cost of the membrane itself. Anyhow, my revision request here is again related to citation. The authors correctly mention two previous papers that also found a drastic reduction in selectivity at higher pressure. However, they did not cite a key recent paper by K. Eum et al (ACS Applied Materials & Interfaces, 8 (38), 25337-25342, 2016) where selectivities of 90+ were maintained at 9 bar. This should be cited/discussed. Similar to Eum's paper, the present authors also find slight reduction in permeance at higher pressures - this is different from the other two papers in which the permeance increased. The explanations for the two types of behavior are different.

Response/action: We agree with the assessment by the reviewer. Hence we further tested the performance of the membranes at a pressure of 9 bar recently. The C_3H_6/C_3H_8 selectivity remained as high as 22.7, which is beyond the prediction of the reviewer and surpasses the upper-bound of polymeric membranes. So far, the C_3H_6/C_3H_8 separation membranes are usually fabricated by polymers (e.g., PIM-1 and 6FDA-DDBT), zeolites, carbons and MOFs. Both C_3H_6 and C_3H_8 can induce plasticization of polymers, and swelling of polymers occurs in pressured feed gases, which leads to increased segmental motion and fractional free

volume (Bachman, J. E. *et al. Nature Mater.* **15**, 845-849 (2016); Koros, W. J. *et al. Nat. Mater.* **16**, 289-297 (2017).). Therefore, the polymeric membranes show higher gas permeability and lower selectivity with higher pressure, especially when the feed pressure exceeds the plasticization pressure (Das, M. *et al. J. Membr. Sci.* **365**, 399-408 (2010); Swaidan, R. J. *et al. J. Membr. Sci.* **492**, 116-122 (2015); Liao, K. S. *et al. J. Membr. Sci.* **515**, 36-44 (2016).). Molecular sieving membranes, including carbon and MOF membranes, are unaffected by plasticization. The decrease of permeance and selectivity in the present study was attributed to two factors. First, more C₃H₈ molecules entered into ZIF-8 pores under higher pressure, which blocked the pores and formed competition, and led to decreased C₃H₆ permeance and C₃H₆/C₃H₈ selectivity (Yu, J. *et al. Chem. Eng. Sci.* **141**, 119-124 (2016); Eum, K. *et al. ACS Appl. Mater. Inter.* **8**, 25337-25342 (2016); Koros, W. J. *et al. Nat. Mater.* **16**, 289-297 (2017).). Second, concentration polarization occurred (Lüdtke, O. *et al. J. Membr. Sci.* **146**, 145-157 (1998); Li, W. *et al. J. Mater. Chem. A* **2**, 2110-2118 (2014).). Because C₃H₆ permeance was much larger than C₃H₈ permeance, the molar ratio of C₃H₆/C₃H₈ at membrane surface was smaller than that of the feed gas mixture. Moreover, the increased amount of permeated C₃H₆ was larger than that of C₃H₈ with increasing pressure. Therefore, the C₃H₆/C₃H₈ ratio at membrane surface decreased with increasing feed pressure, resulting in decreased C₃H₆ permeance and C₃H₆/C₃H₈ selectivity. We have cited the related references including Eum, K. *et al. ACS Appl. Mater. Inter.* **8**, 25337-25342 (2016) and discussed the mechanism of C₃H₆/C₃H₈ separation behavior in the revised manuscript (section: Gas transport behaviours of ZIF-8 membranes).

3) In the cases where the authors have provided additional permeation data (e.g., binary permeation, higher pressures), they should clarify whether these data were collected with the single-fiber membranes or with the 30-fiber modules. It seems the former, but this should be stated in the discussion.

Response/action: We have stated that the additional permeation data were collected with the single-fiber membranes in the revised manuscript.

Reviewer #3

I am partially satisfied with the answers given by the authors. It is OK regarding points 1-7. However, regarding point 8, these membranes even if prepared by a new method show low selectivities (e.g. in case of CO₂/CH₄ in the 6.3-9.5 range) as compared to those in the literature. In addition, this gas separation performance was obtained from single gas permeance measurements and not from mixture separation experiments, as far as I can understand.

Response/action: Previous studies suggested that the most potential applications of ZIF-8 membranes are C₃H₆/C₃H₈ separation and H₂ permselectivity (Liu, Q. *et al. J. Am. Chem. Soc.* 135, 17679-17682 (2013); Brown, A. J. *et al. Science* 345, 72-75 (2014); Liu, Y. *et al. Angew. Chem. Int. Ed.* 54, 3028-3032 (2015); Kwon, H. T. *et al. J. Am. Chem. Soc.* 135, 10763-10768 (2013); Kwon, H. T. *et al. J. Am. Chem. Soc.* 137, 12304-12311 (2015); Eum, K. *et al. Adv. Funct. Mater.* 26, 5011-5018 (2016)). In mixture separation, the prepared membranes showed high permeance in both H₂ permselectivity ($117.8 \times 10^{-7} \text{ mol m}^{-2} \text{ s}^{-1} \text{ Pa}^{-1}$) and C₃H₆/C₃H₈ separation (a C₃H₆ permeance of $2.8 \times 10^{-7} \text{ mol m}^{-2} \text{ s}^{-1} \text{ Pa}^{-1}$), as well as high H₂/C₃H₈ selectivity (3126) and C₃H₆/C₃H₈ selectivity (73.4).

Compared with other membranes, such as polymeric membranes (with CO₂/CH₄ selectivity usually from 20 to 30, and as high as >200, Wang, S. *et al. Energy Environ. Sci.* 9, 3107-3112 (2016)), graphene oxide membranes (with CO₂/CH₄ selectivity of 75, Wang, S. *et al. Energy Environ. Sci.* 9, 1863-1890 (2016)), zeolite membranes (with CO₂/CH₄ selectivity of >100, Li, S. *et al. Adv. Mater.* 18, 2601-2603 (2006); Carreon, M. A. *et al. J. Am. Chem. Soc.* 130, 5412-5413 (2008)) and mixed matrix membranes (with CO₂/CH₄ selectivity usually from 20 to 80, Seoane, B. *et al. Chem. Soc. Rev.* 44, 2421-2454 (2015)), ZIF-8 membranes reported in literatures and present study may not show good CO₂/CH₄ separation performance, because the effective aperture size of ZIF-8 is approximately 0.4 nm (Zhang, C. *et al. J. Phys. Chem. Lett.* 6, 3841-3849 (2015).), which is larger than the kinetic diameters of CO₂ (0.33 nm) and CH₄ (0.38 nm). Even so, our membranes showed a competitive CO₂/CH₄ selectivity compared with other ZIF-8 membranes reported in previous studies (Response Table 1). To focus the separation performance on C₃H₆/C₃H₈ mixture and H₂ permselectivity, we have removed Supplementary Table 2.

The reviewer motioned that the ZIF-8 and ZIF-93 membranes supported by P84 hollow

fibers exhibited high CO₂/CH₄ selectivity of 20.4 and 16.9, respectively (Cacho-Bailo, F. *et al. RSC Adv.* **6**, 5881-5889 (2016); Cacho-Bailo, F. *et al. Chem. Commun.* **51**, 11283-11285 (2015)). The separation performance should be measured by both permeance and selectivity. These two membranes showed small CO₂ permeance of only $0.47 \times 10^{-8} \text{ mol m}^{-2} \text{ s}^{-1} \text{ Pa}^{-1}$ and $0.077 \times 10^{-8} \text{ mol m}^{-2} \text{ s}^{-1} \text{ Pa}^{-1}$, which are three to four orders of magnitude smaller than those of our membranes ($83\text{-}760 \times 10^{-8} \text{ mol m}^{-2} \text{ s}^{-1} \text{ Pa}^{-1}$). Moreover, these two membranes were post-annealed at 175 °C for 24 h to obtain higher selectivity. In fact, the annealed P84 hollow fibers without ZIF layers showed high CO₂/CH₄ selectivity of 15.2, which is slightly less than that of the ZIF/P84 hollow fiber membranes. Apparently, the ZIF layers only improved the selectivity by 5.2 and 1.7, respectively. It is difficult to confirm the real contribution of the ZIF layers in CO₂/CH₄ selectivity.

Response Table 1 | Gas separation properties of the ZIF membranes reported in previous studies for CO₂/N₂, and CO₂/CH₄ systems.

Membrane	Substrate	Thickness (100 nm)	Permeance $\times 10^{-8} \text{ mol m}^{-2} \text{ s}^{-1} \text{ Pa}^{-1}$	Selectivity		Ref
				CO ₂ /N ₂	CO ₂ /CH ₄	
ZIF-8	Al ₂ O ₃	50	2430.0	-	5.1	1*
ZIF-8	Al ₂ O ₃	200	2.1	1.7	4.0	2
ZIF-8/LDH	Al ₂ O ₃	200	3.4	2.4	3.0	3
ZIF-8	Al ₂ O ₃	16	0.4	2.2	2.1	4
ZIF-8	Al ₂ O ₃	25	12.2	2.7	2.9	5
ZIF-8/LDH	Al ₂ O ₃	25	0.8	4.2	12.9	6
ZIF-8/GO	AAO	1	3.44	7.0	7.1	7
ZIF-69	Al ₂ O ₃	400	10	6.3	4.6	8*
ZIF-8	P84	26	0.47	-	20.4	9*
ZIF-93	P84	26	0.077	-	16.9	10*

Note: *mixture gas separation performance

References

- Venna, S. R. & Carreon, M. A. Highly permeable zeolite imidazolate framework-8 membranes for CO₂/CH₄ separation. *J. Am. Chem. Soc.* **132**, 76-78 (2010).

2. Liu, Q., Wang, N., Caro, J. & Huang, A. Bio-inspired polydopamine: A versatile and powerful platform for covalent synthesis of molecular sieve membranes. *J. Am. Chem. Soc.* **135**, 17679-17682 (2013).
3. Liu, Y., Wang, N. Y., Pan, J. H., Steinbach, F. & Caro, J. In-situ synthesis of MOF membranes on ZnAl-CO₃ LDH buffer layer-modified substrates. *J. Am. Chem. Soc.* **136**, 14353-14356 (2014).
4. Shekhah, O. *et al.* The liquid phase epitaxy approach for the successful construction of ultra-thin and defect-free ZIF-8 membranes: pure and mixed gas transport study. *Chem. Commun.* **50**, 2089-2092 (2014).
5. Liu, D., Ma, X., Xi, H. & Lin, Y. S. Gas transport properties and propylene/propane separation characteristics of ZIF-8 membranes. *J. Membr. Sci.* **451**, 85-93 (2014).
6. Liu, Y. *et al.* Remarkably enhanced gas separation by partial self-conversion of a laminated membrane to metal-organic frameworks. *Angew. Chem. Int. Ed.* **54**, 3028-3032 (2015).
7. Hu, Y. X. *et al.* Zeolitic imidazolate framework/graphene oxide hybrid nanosheets as seeds for the growth of ultrathin molecular sieving membranes. *Angew. Chem. Int. Ed.* **55**, 204-2052 (2016).
8. Liu, Y., Zeng, G., Pan, Y. & Lai, Z. Synthesis of highly c-oriented ZIF-69 membranes by secondary growth and their gas permeation properties. *J. Membr. Sci.* **379**, 46-51 (2011).
9. Cacho-Bailo, F. *et al.* MOF-polymer enhanced compatibility: post-annealed zeolite imidazolate framework membranes inside polyimide hollow fibers. *RSC Adv.* **6**, 5881-5889 (2016).
10. Cacho-Bailo, F. *et al.* High selectivity ZIF-93 hollow fiber membranes for gas separation. *Chem. Commun.* **51**, 11283-11285 (2015).